# Nicotine Dependence among Adolescents Single and Dual Cigarette Users

**DOI:** 10.3390/children8020144

**Published:** 2021-02-14

**Authors:** Nawwal Alwani Mohd Radzi, Roslan Saub, Zamros Yuzadi Mohd Yusof, Maznah Dahlui, Sharol Lail Sujak

**Affiliations:** 1Department of Community Oral Health and Clinical Prevention, Faculty of Dentistry, University of Malaya, Kuala Lumpur 50603, Malaysia; nawwal@uitm.edu.my (N.A.M.R.); zamros@um.edu.my (Z.Y.M.Y.); 2Centre of Population Oral Health and Clinical Prevention Studies, Faculty of Dentistry, Universiti Teknologi MARA, Sungai Buloh Campus, Selangor 47000, Malaysia; 3Centre of Population Health, Department of Social and Preventive Medicine, Faculty of Medicine, University of Malaya, Kuala Lumpur 50603, Malaysia; maznahd@ummc.edu.my; 4Negeri Sembilan Oral Health Division, Ministry of Health Malaysia, Seremban, Negeri Sembilan 73000, Malaysia; drsharol@moh.gov.my

**Keywords:** nicotine dependence, HONC, Malaysian, adolescent, dual user, salivary cotinine, smoking

## Abstract

The prevalence of concurrent use of combustible and electronic cigarettes (dual-use) is on the rise among Malaysian adolescents. This study compares nicotine dependence among exclusive cigarette users, e-cigarette users, and dual adolescent users. A total of 227 adolescent smokers completed a self-administrated questionnaire with items based on Hooked on Nicotine Checklist (HONC) incorporated. Endorsement of at least one HONC item indicates nicotine dependence. Exhaled carbon monoxide readings and salivary cotinine data were also collected. Over half (52.9%) of the participants were exclusive e-cigarette users (EC). The prevalence of exclusive conventional cigarette smokers (CC) and dual users was 11.9% and 35.2%, respectively. Adolescents who have mothers with secondary school education were more likely to become addicted to nicotine (Adjusted Odd Ratio (aOR) = 2.72; 95% CI = 1.17–6.32). Adolescents’ “mother’s education” level predicted nicotine dependence. This highlighted the need to target families within the identified demography with a more supportive anti-tobacco program.

## 1. Introduction

Smoking continues to be a leading cause of preventable deaths from noncommunicable diseases around the world. With an estimated 6 million deaths worldwide due to tobacco use, it is the most critical health risk factor for the global population [1]. Cigarette smoking among adolescents is a public health concern as it is linked to a heavier smoking habit as an adult in the future [1,2,3]. 

Compared with adults, adolescents display evidence of addiction at a much lower level of tobacco consumption, which results in stronger addiction, a lower success rate of quitting, and higher mortality rate in later life [2,3]. Approximately 68.4% of Malaysian adolescents begin smoking at the age of 14, with 28.5% exhibiting low nicotine dependence [4]. 

The pattern of tobacco use has shifted from conventional combustible cigarettes to smokeless tobacco and recently, electronic cigarettes (e-cigarettes), with a staggering increase in global sales of e-cigarettes from £0.5 billion in 2009 to £6.1 billion in 2015 [5]. 

Two of the most commonly used e-cigarette devices are the Electronic Nicotine Delivery System and Electronic Non-Nicotine Delivery Systems (ENDS/ENNDS) that vaporize a solution consisting of nicotine, propylene glycol, flavorings, and other chemicals to create an aerosol for inhalation [2]. It can be a very potent vehicle for nicotine delivery, although some countries advocate e-cigarettes as a harm-reduction tool, especially solutions that are claimed to be nicotine-free based on evidence that it is effective in adult’s smoking cessation therapy [1]. E-cigarette use has significant but adverse implication in adolescents, whereby they are still forming their attitude towards smoking [2]. 

The varieties of e-cigarettes available in the market with tempting flavours and sleek, contemporary designs indicate that the tobacco industries understood the importance of strategically positioning and marketing a product that appeals to the younger demographic [6]. This, coupled with an aggressive advertisement campaign on the internet and onshopping malls and other strategic locations with large adolescent catchment, the e-cigarette has become a lifestyle choice among adolescents. Although not backed by scientific evidence, the tobacco industry continues to typically market e-cigarettes as a “healthier” and “cleaner” smoking alternative compared with combustible cigarettes [7].

In terms of local laws and regulations, the adolescents’ accessibility to combustible cigarettes was controlled by a complete ban of sales through legislation to those below 18 years old. Comparatively, the legislation on the sale of e-cigarettes to adolescents is still disjointed. Despite being completely banned in countries such as Australia, Brunei, Japan, and Singapore, the laws in countries such as Malaysia, the Philippines, Vietnam, and United Kingdom allow the selling of e-cigarettes with only sales restriction and regulations [8]. As a result, adults can be seen using an e-cigarette in places where the combustible cigarette was banned, and this will encourage the normalization of e-cigarette usage among the younger generation [6,9,10]. Evidence shows that even among nonsmoking adolescents, e-cigarette use was associated with smoking intention [11,12]. 

The declining international trends of cigarette smoking among adolescents were seemingly replaced by the increasing trends of e-cigarette use in this age group. As e-cigarette use is strongly associated with the use of other tobacco products among youth, particularly combustible tobacco products, a significant number of adolescents were reported to have used both products or become a dual user [1,10]. 

An estimated 9.1% of Malaysian adolescents aged 10 to 19 years old were current e-cigarette users, which accounted for 300,000 adolescents. The prevalence of dual users, however, was 5.2% (170,000 adolescents) [6].

A study showed that adolescents who are dual users are at higher odds of nicotine dependence and have lower cessation intention compared with exclusive cigarette smokers [5]. They are also highly likely to perpetuate intense combustible cigarette use in the future, resulting in heavier nicotine dependence [2]. Studies also linked dual-use with future substance misuse such as with drugs and alcohol [13]. Dual user adolescents were also associated with truancy and poor academic performance in schools [1].

There are two opinions on why adolescents are drawn to e-cigarettes. The first one is that they perceive the e-cigarette as a health-oriented decision that prevents them from smoking a combustible cigarette or other dangerous substance [1]. The other opinion is that it is a form of a rebellious act against the conventional value that brings pleasure and increases positive mood [14]. Adolescents in Malaysia are drawn into experimenting with e-cigarettes due to their taste, smell, and popularity [4].

Because there is limited research on adolescents’ dual-use pattern with nicotine dependence in Malaysia, the objectives of this article are to identify different types of cigarette use among adolescent smokers in Malaysia and the sociodemographical factors that influence their nicotine dependence. The findings of this study will be relevant for regulatory policymakers who are actively trying to understand the association of dual users with nicotine dependence in order to formulate specific policies for preventing nicotine dependence and design appropriate smoking cessation interventions.

## 2. Materials and Methods

### 2.1. Study Design

This cross-sectional study was conducted from October to November 2019.

### 2.2. Sample Population

The sample population was secondary school children who smoked tobacco or e-cigarettes from urban and rural areas in the district of Seremban, Negeri Sembilan, Malaysia. Four schools participated in this study; two from urban and two from rural areas.

### 2.3. Sample Size Calculation

The sample size was calculated using G* Power Statistical Analysis software version 3.1.9.4. Based on the effect size of 0.3 from the previous review of the national school-based smoking prevention program, the calculated sample size was 214 with 30% attrition rate consideration (α = 0.05, power = 0.80) [15].

### 2.4. Sampling Methods

Two levels of sampling were employed; only public non-boarding schools were included. The schools were ranked by their academic achievement before being matched according to the locality. To ensure a uniform school environment, same-gender and vernacular schools were excluded. Eligible schools were matched by their location and academic performance. Data on the previous academic ranking were obtained from the Seremban District’s education office. At the first level, a match-pair school was randomly chosen. At the second level, the participants were purposively recruited based on the inclusion criteria of being a smoker, fit, and healthy.

### 2.5. Instruments

A self-administrated questionnaire was developed to gather information about participants’ demographic data (age, gender, ethnicity, parental education, and parental household income), age of smoking initiation, and self-reported nicotine dependency.

Nicotine dependency was measured using The Hooked on Nicotine Checklist (HONC) [16]. HONC is a 10-item checklist, with the sum of one and above indicated nicotine dependence or loss of autonomy towards nicotine (HONC score ≥1). The English version of the HONC was translated into the Malay language prior to the study.

Exhaled carbon monoxide (CO) level was used to validate the self-reported smoking status of the participants. Those with a CO level of 5 to 6 parts per million (ppm) were light smokers, and ≥7 ppm were frequent smokers [17].

Cotinine is a major metabolite of nicotine with 10–40 h of half-life detectable in saliva, urine, or serum. It provides a reliable means of smoking status of tobacco use over a period of two to three days [18]. Salivary cotinine was collected from the participants as another adjunct to validate self-reporting smoking behavior.

The salivary cotinine of the participants was measured using NicAlert, which is a semiquantitative immunochromatographic assay test strip. It uses a “dipstick” method [18]. The participants’ saliva was collected using an individual plastic tube container. Once the container was filled with saliva, it closed using a snap-on cover. For testing, the salivary tube container was inverted to allow eight drops of saliva to be deposited on the padded end of the NicAlert Strip. The cotinine in the saliva reacted with the chemical in the strip to reveal a coloured band respective to the cotinine concentration in the saliva after approximately 15 to 30 min. Salivary cotinine was then categorized into low salivary cotinine (<10 ng/mL cotinine), medium salivary cotinine (10–100 ng/mL), and high salivary cotinine (100 ng/mL) categories [18].

### 2.6. Data Collection Process

Data were collected as part of a 6 month cluster-randomized control trial of a school-based smoking cessation program. The program consisted of a screening program and three sessions of group briefing to students who smoke, delivered by a government dentist who was also responsible for providing annual dental treatment at the schools. The participants were asked to classify themselves into any of the three groups of cigarette users in the questionnaire: “conventional cigarette” only (CC), “electronic cigarette” only (EC), and dual user [2].

### 2.7. Data Analysis

Descriptive and categorical variables were summarized as frequencies and percentages; chi-square test was used to assess the relationship between sociodemographic factors, different types of cigarette use, and nicotine dependence. Among other variables, simple and multiple logistic regressions were used to identify the factors associated with nicotine dependence in dual users and non-dual users. Nicotine dependence was used as a dichotomous criterion variable (code 0 = not addicted; code 1 = addicted). IBM SPSS v22 (Chicago, IL, USA: SPSS Inc) software was used to analyze the data.

## 3. Results

Overall, 227 adolescent smokers responded. The mean age was 14.62 (SD 1.32). The large majority of the sample were males (94.7%), and 57.7% were from the lower secondary school, which consisted of schoolchildren aged 13 to 15 years (Form 1 to 3). More than half (52.9%) were exclusive e-cigarette users (EC). The majority of the participants went to schools located in the rural area (69.2%). The majority of them were Malays (81.59%).

With regards to the parents’ education level, the majority of fathers (63.4%) and mothers (71.8%) had secondary school education. Most of the adolescents came from households with a monthly income below Malaysian Ringgit (MYR) 3900 (70.9%) (Table 1). 

With regards to the participants’ smoking behavior, the majority of the participants began smoking between the age of 12 and 13 years (41%). More than 81.9% of the participants scored at least 1 point in HONC, indicating they were addicted towards nicotine (loss of autonomy). A small minority (14.5%) of the participants were found to have exhaled carbon monoxide of 5 ppm (and above). Salivary cotinine reading revealed that almost half of them (49.8%) have at least 100 ng/mL of cotinine, indicating frequent smoker.

Table 2 shows the association between sociodemographic characteristics and types of cigarette use among adolescents.

One-third of the CC users were in Form 4 (16 years old) (33.3%), while a little over one-third of the EC users were in Form 5 (17 years old). The small majority of dual users were Form 3 (15 years old) students (36.3%) (*p* = 0.02). Across the academic years, there was an increase trend in the percentage of CC users (Form 1 to Form 4). However, the opposite trend was observed (Form 3 to Form 4) in the EC and dual users, respectively.

Higher percentages of participants with HONC score of one (1) and above were observed for all types of users. Dual users have the highest percentage of 97.5% of those scoring (1) and above, followed by CC user and EC user with 88.9% and 71.4% respectively (*p* = 0.001).

Exhaled breath CO and salivary cotinine were found to be associated with the different types of cigarette use with a *p*-value of 0.001. More than half of the participants in all three types of cigarette user were recorded to have a low level of exhaled CO, with a maximum of 4 ppm of CO in their exhaled breath.

The majority of exclusive CC users were had high salivary cotinine (74.1%), while the majority of EC users had a medium level salivary cotinine (52.5%). Almost two-thirds of dual user smokers were categorised as having a high salivary cotinine level (63.3%).

Simple and multiple logistic regression analyses were performed to identify the factors associated with nicotine dependence. Simple logistic regression provided preliminary results on potential associated factors (*p*-value <0.25) [19]. From this analysis, four variables were found to be potentially significant, that is, type of cigarette user, gender, mother’s education, and CO reading. These variables were included in the multiple logistic regression analysis, and only one variable was found to be significant, which is the mother’s education. The odds of becoming “nicotine dependent” were higher in those whose mothers had education up to secondary school (Adjusted Odd Ratio (aOR) = 2.73; 95% CI = 1.18–6.33) (Table 3).

## 4. Discussion

The objectives of the study were to determine the prevalence of the different types of cigarette user in adolescents and explore how nicotine dependence differed based on their demography, smoking behavior, and types of cigarette use. In this study, we found that the overall prevalence of current e-cigarette users and dual users among a group of secondary schoolchildren smokers in Seremban, Negeri Sembilan was 52.9% and 35.2%, respectively. Both of these percentages were higher than the findings from a similar study conducted in Kuala Lumpur among adolescent smokers (42.2%) [20]. This could be due to the fact that our study involved adolescents from both rural and urban areas, while the Kuala Lumpur study involved adolescents from urban schools only [20]. The increasing trend of CC users and decreasing trend of EC users with age were consistent with findings from prospective studies that show e-cigarette use predicts future conventional cigarette smoking [21,22]. However, caution is needed when interpreting this finding as the sample in the current study was small. A study with a bigger sample size is required to confirm this trend.

A study conducted among Negeri Sembilan’s adolescents revealed that the prevalence of tobacco use in the rural area was higher than in the urban areas [23]. Coupled with lack of implementation, enforcement, and surveillance of tobacco law in Malaysia, access to tobacco product for adolescents in both urban and rural areas is not monitored [24]. Data from the National Tobacco and E-cigarette survey among Malaysian Adolescents (TECMA) revealed that rural adolescents who frequently smoked were more likely to purchase their cigarette from commercial sources such as supermarkets, grocery stores, and road stalls [24].

Evidence from 68 middle-income and low-income countries on tobacco use (inclusive of cigarette, pipe, water pipe, and other smokeless tobacco) showed that the mean prevalence of tobacco use among adolescents was 13.6%, ranging from 2.8% (Tajikistan) to 44.7% (Samoa) [25]. In a study that utilized nationally representative data in Malaysia, the percentages of exclusive cigarette and e-cigarette users were 21.4% and 9.1%, respectively [4]. With regards to the exclusive e-cigarette user, the prevalence was even lower in Hong Kong, with 4.0%, and in Taiwan, with 1.6% [26].

Research regarding dual users among adolescents has been limited, and the lack of a standardized measure complicates comparisons between countries. The percentage of the dual user in this study was 35.2%. In the United States, the percentage of the dual user was 1.7%, while adolescents in Hawaii showed a slightly higher percentage (12%) of dual users [1].

There are discrepancies in our research in terms of the prevalence of the different types of cigarette smokers with the national data and data from other countries. Apart from the larger sample size, the discrepancy can be due to variation in regulation and enforcement measures involving the sales and advertisement of e-cigarettes. In Taiwan, e-cigarettes are prohibited under the Tobacco Hazards Preventive Acts, while Malaysia has regulated e-cigarettes in accordance with the Poison Act 1952, which means that nicotine-containing e-cigarettes would be classified as a regulated pharmaceutical product, despite the fact that sales to below 18 years old will be penalized. In contrast, Hong Kong has implemented a ban on nicotine-containing e-cigarettes [27].

Research shows that chronic exposure to nicotine among adolescents is dangerous as it may induce changes that sensitize the brain to future substance abuse and can result in a depression-like state that requires treatment under antidepressants [2]. This study used HONC to assess potential nicotine dependence among adolescents. There were limited studies that used HONC, particularly on the dual user status of adolescents. In adults, however, a small-scale study among native American dual users revealed the median score of HONC did not differ from that of exclusive cigarette users despite the perception of e-cigarette use to be less harmful than use of conventional cigarettes [28]. The lowest percentage of nicotine addiction in this study was that of the exclusive e-cigarette users (71.4%), and the highest prevalence of nicotine addiction was that of the dual users (97.5%). This could be due to dual users generally having higher nicotine dependence from a concurrent exposure to a different tobacco-related device, which also will lower their intention to quit [11]. A study among Swedish youth associated dual users with having five times the odds of higher HONC score and failed quit attempts compared with exclusive cigarette smokers [29].

Limited research has compared salivary cotinine and the cigarette usage status among adolescents, however, in a study investigating the urinary cotinine concentration in adolescents who used a high-nicotine-content e-cigarette, famously known as “pod”, more signs of nicotine dependence (higher HONC score) were evident compared with exclusive cigarette users [30]. In this research, the relationship between the salivary cotinine and the different variants of e-cigarette devices such as ENDS or ENNDS were not explored because the adolescents could not ascertain whether their e-liquid contains nicotine or not. A study found that 99% of the e-cigarettes sold in the United States contained nicotine [31]. However, there were also cases of e-cigarettes that claimed to be nicotine-free being found to contain nicotine [32]. Findings from a qualitative study also revealed that adolescent smokers were unsure of the nicotine content in the e-cigarettes [33].

A study on salivary cotinine among adolescents in Negeri Sembilan and Kuala Lumpur identified factors that influenced schoolchildren exposure to secondhand smoke, and concluded that Malaysian adolescents have a higher salivary cotinine concentration that can be due to the lack of enforcement of the smoke-free law in Malaysia [34]. In this study, the participants’ CO reading revealed that 85% of them were categorized as having low CO level based on the cut-off point of 4 ppm and below. This was expected because the value was also dependent on the time of the day the measurement was taken using the detection device. Since the CO measurement was collected during their study period in schools, the student might not have a recent combustible cigarette exposure for the day due to smoking restriction at school. Exhaled CO has a relatively short half-life of four hours, which limited the accuracy and reliability in determining the students’ recent smoking exposure [35]. The CO level would be expected to be low in exclusive e-cigarette users, as showed in the current study. This is because only a minimal amount of CO is formed in e-cigarette users as the result of thermal decomposition of the e-liquid materials such as propylene glycol and glycerine [36]. Electronic cigarette aerosols contain carbonyl compounds that were found to be toxic. Although carbonyl and CO were found to be well correlated in the e-cigarette, they were generally less than in conventional cigarettes [36].

Our study showed that maternal education level has a protective factor in the smoking behavior of the adolescents [37,38]. This highlights the importance of taking into consideration parental education and socioeconomic (SES) level in the development stage of a future health promotion program, because low-SES families will require consistent support to prevent their children from smoking [39]. Due to a limited definition of e-cigarettes in this study, there is a need to standardize different generations of e-cigarettes available in the market for future studies [1].

### Limitation

The schools selected in this study are only from Negeri Sembilan state, and despite the similarity in terms of population and SES with a major city like Kuala Lumpur, it has limited generalizability to other states in Malaysia. In this study, 95% of the sample were male adolescents. This could be because male adolescents were more likely to confess that they were smokers than female adolescents; a trend that was observed in a nationally representative sample [40]. Furthermore, Malaysian male adolescents were more likely to believe that smoking is socially acceptable and were less reluctant to hide their smoking status [41]. A similar finding was reflected in another national survey, where 25.3% of males were reported to smoke compared to 6.7% females [40].

The participants’ self-reported information regarding their smoking behavior could also be influenced by the social norms of their friends in school. Their daily usage frequency and e-cigarette topography data would provide a better understanding of the adolescents’ nicotine dependence.

## 5. Conclusions

More than half of the adolescents were exclusive e-cigarette users, and their nicotine dependence status was found to be associated with the mother’s education level. This is an essential finding in the planning of a future program that could further protect the “at-risk” population.

## Figures and Tables

**Table 1 children-08-00144-t001:** Sociodemographic characteristics of the study participants.

Characteristic	*n* (%)
Age (years)	Mean (SD) 14.62 (1.32)
Gender:	
Male	215 (94.7)
Female	12 (5.3)
Class:	
Form 1 (13 years old)	28 (12.3)
Form 2 (14 years old)	47 (20.7)
Form 3 (15 years old)	56 (24.7)
Form 4 (16 years old)	38 (16.7)
Form 5 (17 years old)	58 (25.6)
Smoking status:	
Conventional Cigarette (CC)	27 (11.9)
E-cigarette only (EC)	120 (52.9)
Dual user	80 (35.2)
School Location:	
Urban	70 (30.8)
Rural	157 (69.2)
Ethnicity:	
Malay	185 (81.5)
Chinese	5 (2.2)
Indian	34 (15.0)
Others	3 (1.3)
Father’s Education:	
Primary school	23 (10.1)
Secondary school	144 (63.4)
University	53 (23.3)
Mother’s Education:	
Primary school	13 (5.7)
Secondary School	163 (71.8)
University	46 (20.3)
Household Income (MYR) *:	
≤3900	161 (70.9)
3900–8300	49 (21.6)
≥8300	11 (4.8)
Initiation age of smoking (years):	
≤7	9 (4.0)
8–9	13 (5.7)
10–11	31 (13.7)
12–13	93 (41.0)
14–15	61 (26.9)
≥16	15 (6.6)
HONC score:	
Not Nicotine Dependent (score <1)	39 (17.2)
Nicotine Dependent (score ≥1)	186 (81.9)
Carbon monoxide (ppm):	
Low (0–4 ppm)	193 (85.0)
High (≥5 ppm)	33(14.5)
Salivary cotinine (ng/mL):	
Low (<10 ng/mL)	25 (11.0)
Medium (10–100 ng/mL)	88 (38.8)
High (>100 ng/mL)	113 (49.8)

SD = standard deviation, ppm = parts per million, HONC = Hooked on Nicotine Checklist, MYR = Malaysian Ringgit (MYR 4.08 = 1 USD).

**Table 2 children-08-00144-t002:** Association between sociodemographic characteristics and types of cigarette user (*N* = 227).

Characteristic	CC Only (*n* = 27) *n* (%)	EC Only (*n* = 120) *n* (%)	Dual User (*n* = 80) *n* (%)	*p*-Value
Gender				0.244 ^b^
Male	100 (27)	111 (92.5)	77 (96.3)	
Female	0 (0)	9 (7.2)	3 (3.8)	
Class				0.002 *^,a^
Form 1 (13 years old)	2 (7.4)	20 (16.7)	6 (7.5)	
Form 2 (14 years old)	5 (18.5)	23 (19.2)	19 (23.8)	
Form 3 (15 years old)	6 (22.2)	21 (17.5)	29 (36.3)	
Form 4 (16 years old)	9 (33.3)	15 (12.5)	14 (17.5)	
Form 5 (17 years old)	5 (18.5)	41 (34.2)	12 (15)	
School location				0.057 ^a^
Urban	3 (11.1)	39 (32.5)	28 (35.0)	
Rural	24 (88.9)	81 (67.8)	52 (65.0)	
Ethnicity				0.083 ^b^
Malay	22 (81.5)	104 (86.7)	59 (73.8)	
Chinese	1 (3.7)	1 (0.8)	3 (3.8)	
Indian	3 (11.1)	15 (12.5)	16 (20.0)	
Others	1 (3.7)	0 (0)	2 (2.5)	
Father’s Education				0.344 ^a^
Primary school	5 (19.2)	9 (7.8)	9 (11.4)	
Secondary school	17 (65.4)	79 (68.7)	48 (60.8)	
University	4 (15.4)	27 (23.5)	22 (27.8)	
Mother’s Education				0.748 ^b^
Primary school	1 (3.8)	5 (4.2)	7 (9.0)	
Secondary school	20 (76.9)	88 (74.6)	55 (70.5)	
University	5 (19.2)	25 (21.2)	16 (20.5)	
Household Income (MYR) *				0.153 ^b^
≤3900	17 (68)	84 (71.2)	60 (76.9)	
3900–8300	8 (32)	29 (24.6)	12 (15.4)	
≥8300	0 (0)	5 (4.2)	6 (7.7)	
Initiation age of smoking (years)				0.127 ^b^
≤7	1 (3.7)	2 (1.7)	6 (7.7)	
8–9	1 (3.7)	8 (6.8)	4 (5.1)	
10–11	3 (11.1)	19 (16.2)	9 (11.5)	
12–13	17 (63.0)	41 (35)	35 (44.9)	
14–15	3 (11.1)	37 (31.6)	21 (26.9)	
≥16	2 (7.4)	10 (8.5)	3 (3.8)	
HONC score				<0.001 *^,a^
Not Nicotine Dependent (score < 1)	3 (11.1)	34 (28.6)	2 (2.5)	
Nicotine Dependent (score ≥ 1)	24 (88.9)	85 (71.4)	77 (97.5)	
CO ppm				<0.001 *^,a^
Low (0–4ppm)	16 (59.3)	118 (98.3)	59 (74.7)	
High (≥5ppm)	11 (40.7)	2 (1.7)	20 (25.3)	
Salivary cotinine				<0.001 *^,a^
Low (<10 ng/mL)	1 (3.7)	14 (11.7)	10 (12.7)	
Medium (10–100 ng/mL)	6 (22.2)	63 (52.5)	19 (24.1)	
High (>100 ng/mL)	20 (74.1)	43 (35.8)	50 (63.3)	

SD = standard deviation, ppm = parts per million, HONC = Hooked on Nicotine Checklist, MYR = Malaysian Ringgit (MYR 4.08 = 1 USD). ^a^ Chi-square test, ^b^ Fisher’s exact test, * *p*-value <0.05.

**Table 3 children-08-00144-t003:** Factors associated with nicotine dependence among different types of smoking classification of adolescent smokers (addicted, 1; not addicted, 0).

Variable	Simple Logistic Regression		Multiple Logistic Regression	
	Crude OR (95% CI)	*p*-Value	Adjusted OR (95 CI%)	*p*-Value
Type of cigarette smoker				
Conventional cigg. (CC) *	1	<0.001	1	
E-Cigarette only (EC)	0.31 (0.08,1.10)		0.48 (0.12,1.89)	0.004
Dual User	4.81 (0.75,30.51)		5.94 (0.90,39.37)	
Age	0.90 (0.69,1.17)	0.419		
Gender				
Female *	1		1	
Male	2.54 (0.54,2.36)	0.145	2.08 (0.53,8.11)	0.294
School Location				
Urban *	1			
Rural	1.13 (0.54,2.36)	0.742		
Ethnicity		0.82		
Malay *	1			
Chinese	0.91 (0.10,8.42)			
Indian	1.71 (0.57,5.18)			
Father’s Education				
Degree *	1	0.511		
Primary/Secondary	1.31 (0.58,2.74)			
Mother’s Education				
Degree *	1	0.021	1	0.019
Primary/Secondary	2.47 (1.14,5.36)		2.73 (1.18,6.33)	
Household Income				
(MYR)				
≥8300 *	1			
<8300	1.10 (0.22,5.33)	0.901		
Carbon monoxide (ppm)				
Low (0–4 ppm) *	1		1	
High (≥5 ppm)	7.95 (1.05,60.02)	0.044	3.03 (0.34,26.83)	0.32
Salivary Cotinine (ng/mL)				
Low (<99 ng/mL) *	1			
High (>100 ng/mL)	0.62 (0.18,2.18)	0.453		
Initiation age of smoking				
After age 10 *	1			
Before age 10	0.96 (0.31,3.01)	0.945		

* = reference group, SD = standard deviation, ppm = parts per million, HONC = Hooked on Nicotine Checklist, MYR = Malaysian Ringgit (MYR 4.08 = 1 USD). *p*-value < 0.05.

## Data Availability

Data available on request due to restrictions from the funding agency.

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
