# Peer review of "Nicotine Dependence among Adolescents Single and Dual Cigarette Users"

_children, 2021, doi:10.3390/children8020144_

Round 1

Reviewer 1 Report

For Authors,

I have made some comments. Please check the attached document for details.

Reviewer 2 Report

This study examined dual-use and exclusive use of cigarettes and e-cigarettes among high school students who either vaped or smoked - the students were from one state in Malaysia. The study also examined factors that influence a binary measure of nicotine dependance. The presentation of the methods and results are clear. A few adits and revisions would improve the paper.

  1. Line 75 - the statement about drugs and alcohol needs a reference
  2. Line 92 - the sample population needs to state that only students who vaped or smoked were in the sample
  3. The limitations should address why 95% of the sample were males
  4. Page 5 - the finding that the percentage of students exclusively smoke cigarettes increased as form increased is interesting and should be addressed in the Discussion. The finding is consistent with prospective findings that e-cigarette use predicts future cigarette smoking.
  5. Line 190 - Regarding CO levels - the authors should address why CO would be expected to be very low among exclusive e-cigarette users

Round 2

Reviewer 2 Report

The authors have addressed my comments about the content of the manuscript.